# Amantadine inhibits known and novel ion channels encoded by SARS-CoV-2 in vitro

Trine Lisberg Toft-Bertelsen[1,2,6], Mads Gravers Jeppesen[1,3,6], Eva Tzortzini [4], Kai Xue[5], Karin Giller[5], Stefan Becker [5], Amer Mujezinovic[1], Bo Hjorth Bentzen[1], Loren B. Andreas [5], Antonios Kolocouris[4], Thomas Nitschke Kledal [3✉] & Mette Marie Rosenkilde [1✉]

The dire need for COVID-19 treatments has inspired strategies of repurposing approved drugs. Amantadine has been suggested as a candidate, and cellular as well as clinical studies have indicated beneficial effects of this drug. We demonstrate that amantadine and hexamethylene-amiloride (HMA), but not rimantadine, block the ion channel activity of Protein E from SARS-CoV-2, a conserved viroporin among coronaviruses. These findings agree with their binding to Protein E as evaluated by solution NMR and molecular dynamics simulations. Moreover, we identify two novel viroporins of SARS-CoV-2; ORF7b and ORF10, by showing ion channel activity in a *X. laevis* oocyte expression system. Notably, amantadine also blocks the ion channel activity of ORF10, thereby providing two ion channel targets in SARS-CoV-2 for amantadine treatment in COVID-19 patients. A screen of known viroporin inhibitors on Protein E, ORF7b, ORF10 and Protein 3a from SARS-CoV-2 revealed inhibition of Protein E and ORF7b by emodin and xanthene, the latter also blocking Protein 3a. This illustrates a general potential of well-known ion channel blockers against SARS-CoV-2 and specifically a dual molecular basis for the promising effects of amantadine in COVID-19 treatment.

---

[1] Department of Biomedical Sciences, Faculty of Health and Medical Sciences, University of Copenhagen, Copenhagen, Denmark. [2] Department of Neuroscience, Faculty of Health and Medical Sciences, University of Copenhagen, Copenhagen, Denmark. [3] Synklino ApS, Charlottenlund, Denmark. [4] Laboratory of Medicinal Chemistry, Section of Pharmaceutical Chemistry, Department of Pharmacy, National and Kapodistrian University of Athens, Panepistimioupolis-Zografou, Athens, Greece. [5] Department of NMR-based structural biology, Max Planck Institute for Biophysical Chemistry, Göttingen, Germany. [6] These authors contributed equally: Trine Lisberg Toft-Bertelsen, Mads Gravers Jeppesen. ✉email: tnk@synklino.com; rosenkilde@sund.ku.dk

The ion channel blocker amantadine has been used for >45 years in the clinic for the treatment of influenza A infections[1] and Parkinson's disease[2]. A recent retrospective cohort study evaluating amantadine amongst other antivirals did not find any significant benefit of amantadine in the treatment of coronavirus disease 2019 (COVID-19) patients[3]. However, a study based on self-reported COVID-19 disease among users of amantadine for neurological diseases[4] and a small-scale treatment of COVID-19 patients with amantadine[5] suggested a positive impact. Moreover, amantadine was recently shown to inhibit Severe acute respiratory syndrome coronavirus 2 (SARS-CoV-2) replication in Vero E6 cells[6].

Ion channels are important drug targets as exemplified by multiple drugs controlling the cardiovascular system, locomotion, and various functions in the central nervous system[7]. In their genomes, viruses may encode ion channels, denoted viroporins that are formed by oligomerization of transmembrane units[8]. Over the past decades, an increasing number of both cation- and anion-conducting viroporins have been identified and proposed to play central roles in the viral life cycle, in addition to having a huge impact on pathologies in the host[8]. Viroporins have been identified in a vast number of pathogenic viruses including hepatitis C (HCV), HIV-1, and influenza A viruses as well as picornaviruses and coronaviruses[8]. Several inhibitors of the ion channel activity of these viroporins[8] have been identified. This includes marketed drugs, such as amantadine as well as rimantadine, which both target the M2 viroporin in influenza A virus[9–11] and hexamethylene-amiloride (HMA) that blocks influenza A M2 and SARS-CoV-1 Protein E[12,13]. Amantadine has been observed to also block the SARS-CoV-1 Protein E viroporin[14].

SARS-CoV-2 is the cause of the ongoing pandemic of COVID-19. It is highly homologous to the deadly SARS-CoV-1 (also known as SARS-CoV), giving rise to the "SARS" epidemic in 2002, and to the Middle East respiratory syndrome coronavirus giving rise to MERS in 2012[15]. One conserved viroporin has been identified in all of these three coronaviruses, the homo-pentameric cation-conducting Protein E[16]. In contrast, the open reading frame (ORF) Protein 3a, which is a homodimeric, cation-conducting ion channel, is found only in SARS-CoV-1 and −2[17]. In SARS-CoV-1, expression of both viroporins promotes virus replication and virulence[18] and deletion of the Protein E gene attenuates the virus, resulting in faster recovery and improved survival in infected mice[19]. At the cellular level, deletion of Protein E decreases edema accumulation, the major determinant of the deadly acute respiratory distress syndrome, in addition to reducing levels of inflammasome-activated IL-1b, indicating that Protein E ion channel function is required for inflammasome activation[19]. Hence, Protein E ion channel activity represents a determinant for SARS-CoV-1 virulence, which mirrors the pathology associated with the severe cases of SARS-CoV-2 infection. This suggests that inhibition of the SARS-CoV-2 Protein E viroporin might likewise limit pathogenicity and thus be of therapeutic value in SARS-CoV-2 infections.

SARS-CoV-1 Protein 3a has been reported to form an emodin-sensitive K[+]-permeable cation channel[17] and to be implicated in inflammasome activation as well as both apoptotic and necrotic cell death[20], whereas SARS-CoV-2 Protein 3a has been implicated in apoptosis and inhibition of autophagy in vitro[21]. In mouse models of SARS-CoV-1 infection, genomic deletion of Protein 3a reduced viral titer and morbidity[18]. Protein 3a has therefore been considered a potential target for vaccines or therapeutics to treat SARS[22–24]. Still, the precise role of Protein 3a in disease pathogenesis is unclear, precluded in part by the lack of a mechanistic understanding, but recently cryo-electron microscopy, electrophysiology, and fluorescent ion flux assays have begun to elucidate on SARS-CoV-2 Protein 3a structure and function[25].

In contrast to Protein E and 3a, much less is known about the function and structure of the proteins denoted ORF7b and ORF10, yet they both encode potential transmembrane helixes. ORF7b is a relatively uncharacterized accessory protein of SARS-CoV-2, yet with a relatively high expression level[26]. In SARS-CoV-1, it has been found to localize to the Golgi compartment as a membrane integral protein and to be incorporated into mature viral particles[27,28]. ORF10, found in SARS-CoV-2, but not in SARS-CoV-1, has not yet been assigned any function and has been suggested both as a non-functional gene[29] and as a functional gene undergoing positive selection[30].

Here, we used electrophysiology in Xenopus laevis (X. laevis) oocytes to explore the ion channel activity of SARS-CoV-2 Protein E and describe that both amantadine and HMA block its activity, while rimantadine is inactive. By using solution nuclear magnetic resonance (NMR) spectroscopy in micelles and molecular dynamics (MD) simulations, we explored the binding properties of amantadine, rimantadine, and HMA to Protein E. We also demonstrated ion channel activity of protein 3a, ORF7b and ORF10, and found that all four ion channels can be inhibited by at least one drug among a selection of known viroporin inhibitors.

## Results

**Amantadine and other inhibitors block the ion channel function of SARS-CoV-2 Protein E.** Amantadine has previously been shown by surface plasmon resonance to bind to the native transmembrane α-helical region of the SARS-CoV-1 Protein E (ETM) and by electrophysiology to block full-length Protein E (aa 1–75)-mediated conductance at low mM concentration[12,14]. It has also been shown using isothermal titration calorimetry to bind the native transmembrane α-helical region of influenza A M2 Protein and by electrophysiology to block the full-length Protein M2 (aa 1–98) at low μM concentration[31]. Intriguingly, the transmembrane region of Protein E from SARS-CoV-2 is 100% identical to that of SARS-CoV-1. We, therefore, decided to probe if amantadine blocks Protein E from SARS-CoV-2. To establish ion channel function, we employed the X. laevis oocyte expression system and monitored current activity with a conventional two-electrode voltage clamp (TEVC). During our studies, it was published, using solid-state NMR (ssNMR) of the ETM from SARS-CoV-2 that both amantadine and HMA bind the ETM pore[32]. Moreover, the amantadine derivative memantine was suggested to block SARS-CoV-2 replication based on a bacterial assay[33]. We found that when expressed in oocytes, full-length Protein E from both SARS-CoV-2 and -1 presented a significantly augmented current activity as compared to control oocytes indicating ion channel activity (Fig. 1a and Supplementary Fig. 1a, respectively). Amantadine efficiently blocked the activity of SARS-CoV-2 Protein E (77%; $p = 0.006$, Fig. 1b), and consistent with previous data[14], also blocked that of SARS-CoV-1 Protein E (66%, $p = 0.005$, Supplementary Fig. 1b)[12,14].

Besides amantadine, several other marketed and experimental drugs have been described as inhibitors of viroporins. To determine whether SARS-CoV-2 Protein E ion channel activity could be blocked by any of these drugs, we selected seven (rimantadine, adamantane, HMA, emodin, xanthene, pyronin B, and -Y), and monitored the ion channel activity in the presence of 10 μM of each drug (Fig. 1c–i). Among these, HMA blocked the ion channel activity (50%, $p = 0.0094$, Fig. 1e) consistent with its established binding to the transmembrane region of SARS-CoV-2 Protein E[12,14]. A similar blocking was observed for emodin (60%, $p = 0.0011$, Fig. 1f) and xanthene (80%, $p = 0.0007$, Fig. 1g). In contrast, rimantadine, adamantane, and pyronin B and Y did not affect the ion channel activity (Fig. 1c, d, h, i). The lack of

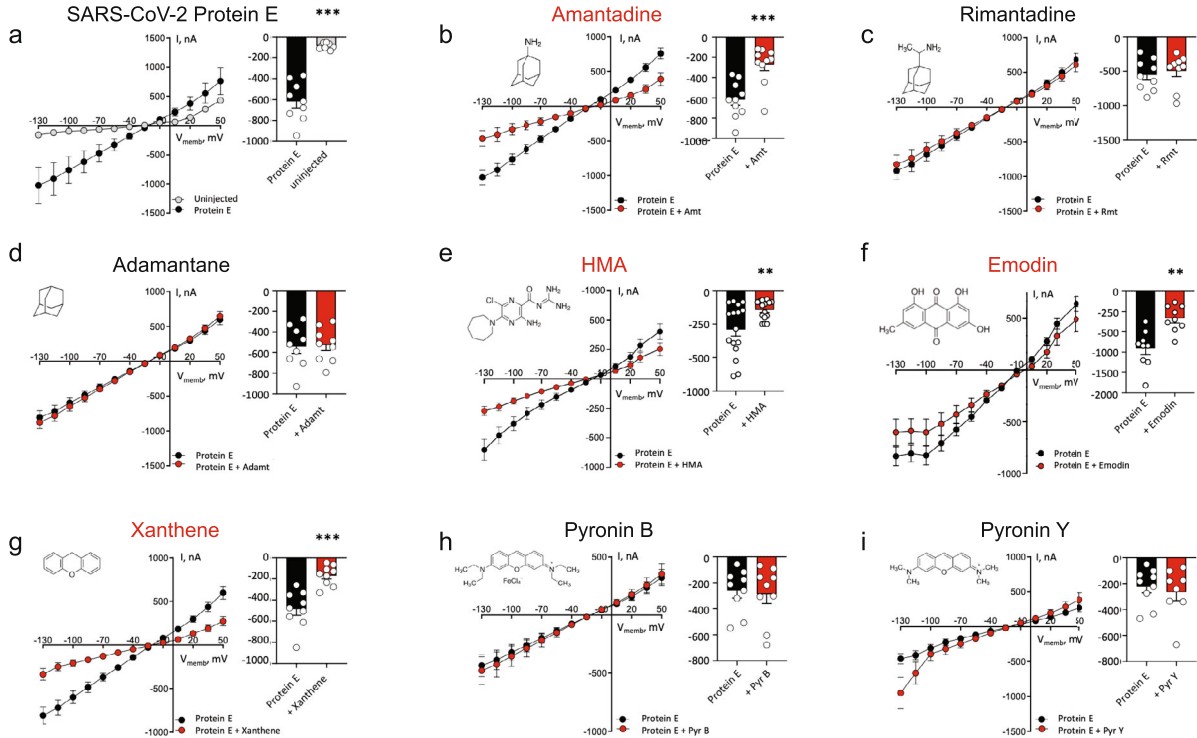

**Fig. 1 Amantadine among other inhibitors blocks the ion channel function of SARS-CoV-2 Protein E. a** Summarized and averaged I/V relations in SARS-CoV-2 Protein E-expressing oocytes revealed significantly different current activity compared to control (uninjected) oocytes. The current activity at −85 mV in SARS-CoV-2 Protein E-expressing oocytes normalized to that of uninjected oocytes. **b–i** Eight compounds (10 μM) were tested on SARS-CoV-2 Protein E activity. Amantadine (**b**), HMA (**e**), emodin (**f**), and xanthene (**g**) blocked the activity, whereas no inhibition of SARS-CoV-2 Protein E by rimantadine (**c**), adamantane (**d**), pyronin B (**h**), or pyronin Y (**i**) was observed. **b–i** current activity at −85 mV with the treatment of the specific drugs normalized to the current activity obtained without treatment. Statistical significance was determined with unpaired Student $t$ test, $**P < 0.01$; $***P < 0.001$, of $n = 4$ biologically independent experiments.

inhibition by rimantadine was surprising, given its previously described binding to and inhibition of Protein E from SARS-CoV-1[34] and the amino-acid identity in the transmembrane regions of Protein E from SARS-CoV-1 and -2 (Supplementary Fig. 2a and b). We, therefore, probed the activity of rimantadine and the three other inhibitors adamantane, HMA, and emodin on this viroporin, and consistent with previous data, rimantadine did indeed block the ion channel activity of SARS-CoV-1 (37%, $p = 0.013$, Supplementary Fig. 1c). The same pattern was observed for adamantane that also blocked Protein E from SARS-CoV-1, but not that of SARS-CoV-2 (Supplementary Fig. 1d and Fig. 1d, respectively). In contrast, HMA and emodin blocked the Protein E ion channels from both SARS-CoV-1 and -2 (Supplementary Fig. 1e and f and Fig. 1e and f). To further explore the drug interaction with SARS-CoV-2 Protein E, we recorded solution NMR data in micelles and performed molecular dynamics simulations.

**Solution NMR reveals that amantadine, HMA, and rimantadine interact differently with Protein E.** Protein E forms an α-helical transmembrane bundle of five protein monomers[16]. However, there is variation in the suggested length of the transmembrane domain and the α-helical bundle tilt, depending on the Protein E construct explored (SARS-CoV-1 or -2), as well as the lipid or detergent used for reconstitution. In a solution NMR study of SARS-CoV-1 Protein E (aa 8–65) in dodecylphosphocholine (DPC) or lyso-myristoylphosphatidylglycerol (LMPG) micelles (PDB ID 5X29[16]), a structure was suggested

with a transmembrane domain formed by a 24-residue α-helix (aa 14–37) connected to a broken cytoplasmic α-helix (aa 42–64) through an unstructured linker. The transmembrane region of Protein E (aa 8–38) from SARS-CoV-2, investigated using ssNMR in dimyristoylphosphatidylcholine (DMPC)/dimyristoylpho-sphatidyl-glycerol bilayers (PDB ID 7K3G[32]) was reported to form a transmembrane α-helix of similar length (aa 14–34), but with a narrower pore at the center of the pentameric bundle. A recent solution NMR study of full-length Protein E (aa 1–75) from SARS-CoV-2 in hexadecylphosphocholine micelles suggests that after seven unstructured residues (aa 1–7), a 36-residue transmembrane domain is formed (aa 8–43), which is connected through an unstructured 10-residue fragment (aa 44–52) to the nine-residue cytoplasmic α-helix (aa 53–61), again followed by unstructured cytoplasmic residues (aa 61–75)[35,36]. The binding area of amantadine in Protein E has previously been suggested to include transmembrane as well as C-terminal residues, according to the solution NMR analysis for Protein E (aa 8–65) of SARS-CoV-1 in micelles[16], and ssNMR of Protein E (aa 8–38) of SARS-CoV-2 in complex with amantadine in lipid bilayers[32]. HMA, which efficiently blocked Protein E from both SARS-CoV-2 and -1 (Fig. 1e and Supplementary Fig. 1e) was shown in one study to bind to the same region as amantadine in Protein E from SARS-CoV-1[16], however in another study to bind between aa 6 and 18 according to solution NMR in micelles of the full-length Protein E (aa 1–75) in HPC micelles[36]. To further explore the sequence-specific effects of amantadine, rimantadine, and HMA, we employed solution NMR of the full-length SARS-CoV-2 protein E (aa 1–75) in n-hexadecylphosphocholine (Fos-Choline-16)

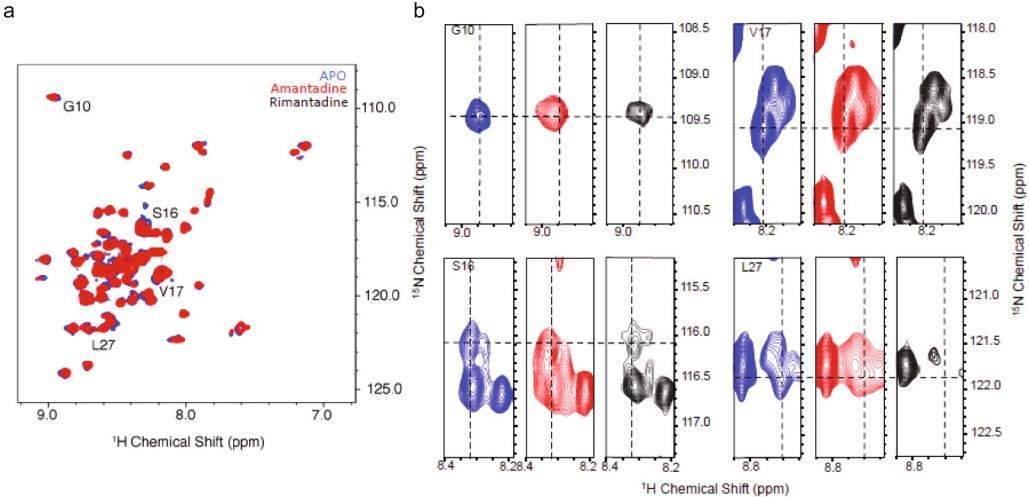

**Fig. 2 CSPs from ¹H-¹⁵N HSQC spectra of full-length Protein E from SARS-CoV-2 in the apo form and after adding amantadine (red) or rimantadine (black). a** The backbone amide region of the spectrum, with notable sequence-specific resonance assignments indicated. **b** Expansion of the spectrum showing shift perturbations for the residues G10, S16, V17, and L27. Measurements were recorded at 50 °C using a Bruker Prodigy probe on a 600 MHz instrument.

micelles (recall that rimantadine blocked Protein E from SARS-CoV-1, but not that of SARS-CoV-2 (Fig. 1c and Supplementary Fig. 1c)). The three drugs were added in 10-fold excess compared to Protein E, i.e., at 2 mM concentrations. The assignment of full-length Protein E from SARS-CoV-2 in micelles was performed using standard transverse relaxation optimized spectroscopy (TROSY) based triple resonance pulse sequences to link three types of atomic nuclei, ¹H, ¹⁵N, and ¹³C. TROSY allows for studies of large molecules or complexes. The sequences HNCA HNCOCA, HNCO, HNCACO, and HNCACB[37] were employed and the resulting assignment was validated against the recent reports of full-length Protein E[36]. The backbone chemical shifts are similar in the transmembrane region, and differ near the Cys to Ala mutations of residues 40, 43, and 44. Figure 2 shows the heteronuclear single quantum coherence spectrum, which is used to determine proton–nitrogen single bond correlations, and separating signals in order to measure the chemical shift perturbations (CSPs) upon ligand binding to Protein E for amantadine and rimantadine. Per-residue CSP values for all three compounds (HMA, amantadine, and rimantadine) are shown in Supplementary Fig. 3a-c as calculated according to $\sqrt{((\delta\_H^2 + 0.14*\delta\_N^2)/2)}$. For both amantadine and HMA, isolated residues near the N-terminal binding site are similarly perturbed, particularly for the two residues following the polar pore residue N15. For residues G10 and S16, the CSPs occur in a consistent direction for the two compounds, while the V17 resonance moves in opposite directions. For rimantadine shift perturbations are minor for S16 and V17, but a significant CSP is observed at L27. CSPs cannot be used to unequivocally pinpoint the location of binding, but often do highlight the site of direct interaction. The largest CSPs were observed for amantadine and HMA at residues 16 and 17, which is consistent with the binding poses suggested in MD simulations (see below). These two drugs, and particularly HMA, may interact also with the C-terminus of the transmembrane region as shown from the CSPs for residues 40–42, 45–50, 60–70. Generally, less significant CSPs were detected for rimantadine suggesting a weaker interaction with Protein E.

**Molecular dynamics simulations support differential interaction of amantadine, rimantadine, and HMA with Protein E.** To investigate possible binding profiles of the drugs to Protein E, we performed restrained 100 ns-MD simulations of amantadine, rimantadine, and HMA in complex with Protein E (aa 8–65) (PDB ID 5X29[16]) in 1-palmitoyl-oleoyl-sn-glycero-phosphocholine (POPC) lipid bilayer (see Supporting Information). This approach was chosen since the unrestrained MD simulations of the experimental Protein E structure (PDB ID 5X9[16]) resulted in a significant protein unfolding. We chose to use the experimental structure of Protein E (aa 8–65) (PDB ID 5X29[16]) resolved by solution NMR since it includes most of the residues present in the full-length protein investigated here by electrophysiology and solution NMR in micelles. We observed that the outward orientation of amantadine or HMA is stable inside the Protein E pore (Fig. 3) in contrast to an inward orientation (Supplementary Fig. 4a and b). In the outward orientation the ammonium group or guanidinium group, respectively, was oriented towards the N-terminus and formed hydrogen bonds with the amide side chain of N15. The adamantyl or azepinyl group of amantadine and HMA, respectively, were oriented towards the C-terminus in contact with L18 isobutyl side chains. Due to the wide pore of this structure, the drug positioning was flexible and also occasionally moved deeper in the C-terminus in agreement with the CSPs in residues at the C-terminus observed in our and others[36] solution NMR studies of the full-length Protein E. The 100 ns restrained MD simulations of rimantadine inside the pore of Protein E (aa 8–65) showed that the drug is moving between the inward and outward orientation (Supplementary Fig. 4c and d) producing a significant perturbation of the Protein E with a high RMSD ~ 5 Å of the Cα carbons. These results suggest an unstable binding of rimantadine likely due to the repulsions between the lipophilic CHCH₃ adduct in rimantadine, compared to amantadine, with the surrounding polar side chains of N15 and T11. These observations agree with our solution NMR results suggesting a weaker interaction of rimantadine with Protein E (aa 1–75) compared to amantadine or HMA.

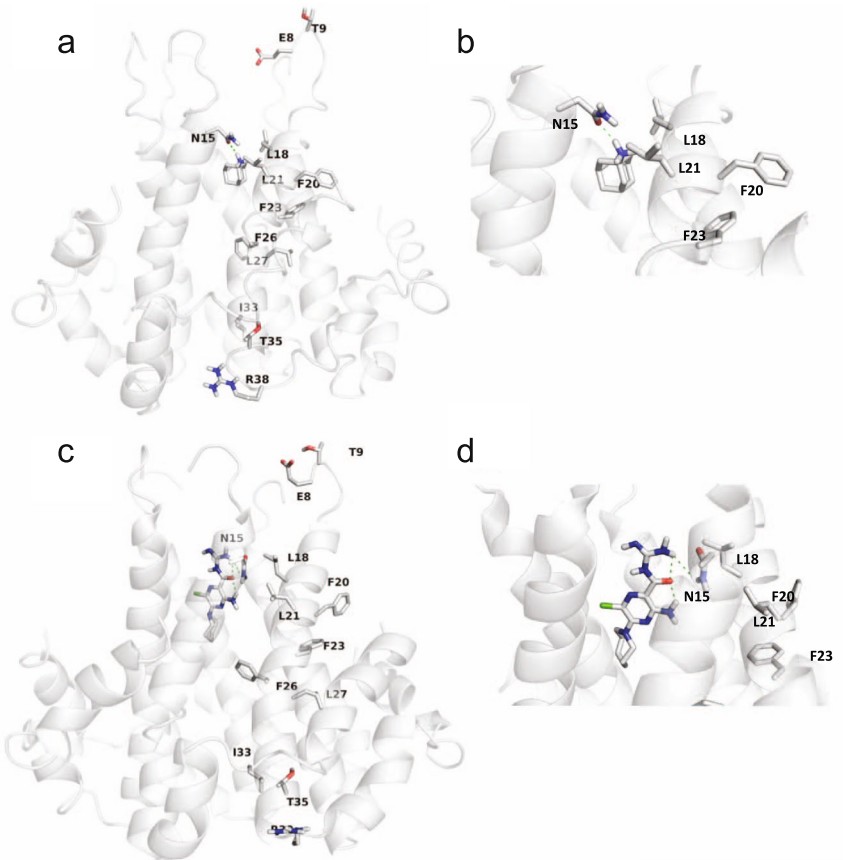

**Fig. 3 MD simulations models of Protein E (aa 8–65) in complex with amantadine and HMA. a** Complex with amantadine, with inset in (**b**) zooming in at the binding site. **c** Complex with HMA, with inset in (**d**) zooming in at the binding site. The models depict an outward orientation of the drugs with Protein E in an open state (ligand carbons and Protein E in gray) from restrained 100 ns-MD simulations with OPLS2005 force field and a force constant of 2 kcal mol Å$^{-2}$ to the Ca atoms of Protein E. Ligand and amino–acid residues (shown in only one of the five α-helices for clarity) are shown as sticks. The PDB ID 5X29[16] was used as starting structure (aa 8–65) for the Protein E-amantadine complex.

**SARS-CoV-2 ORF7b and ORF10 function as ion channels.**
Viroporins have been identified in multiple viruses, in some cases more than one per virus[8]. As a search for novel drug targets among virus-encoded ion channels, we looked in the genome of SARS-CoV-2 and identified two potential novel viroporins; ORF7b and ORF10 (Fig. 4a). ORF7b is a small protein composed of 43 residues with ~85% sequence identity between the SARS-CoV-1 and -2 homologs, while the SARS-CoV-2 and bat coronavirus RaTG13 proteins share 97% sequence identity[38]. In SARS-CoV-1, ORF7b has been identified as a transmembrane protein with an external N-terminus and cytoplasmic C-terminus[39]. Moreover, ORF7b from SARS-CoV-2 was recently suggested to assemble into a pentamer and to be involved in heart arrhythmia and loss of smell through interactions with other proteins[40]. We discovered amino-acid sequence homology of the core of ORF7b to Protein E from SARS-CoV-1 and -2 in a consensus region containing three conserved Phe residues (Supplementary Fig. 2a and b) all facing the lipid membrane, among which F23 and F26 are involved in helix-helix interactions in the homo-pentameric bundle[32]. Moreover, SARS-CoV-2 ORF7b has a short amino-acid sequence motif at the C-terminus of its predicted transmembrane region, which is highly homologous to a sequence at the N-terminus of the transmembrane region of Protein E of transmissible gastroenteritis virus (Supplementary Fig. 2a). There is an additional homology between SARS-CoV-1 ORF7b and ORF8a in their C-termini (Supplementary Fig. 2b).

SARS-CoV-2 ORF10 is a suggested 38 amino-acid long protein, the gene of which is positioned at the 3′-end of the viral genome. It is also present in Pangolin-CoV (with ~97% sequence identity) and in a truncated version in bat-, civit, and SARS-CoV-1 genomes[30,41,42]. The truncated versions have only the first 29 amino acids, which leave the predicted transmembrane region intact and thereby the protein potentially functionally intact[43].

To explore the ion channel function of ORF7b and ORF10, we again employed the *X. laevis* oocyte expression system and monitored current activity with conventional TEVC. When expressed in oocytes, ORF7b and ORF10 presented significantly augmented current activity to the same extent as Protein E as compared to control oocytes indicating ion channel activity of both (Fig. 4b). Consistent with previous data[17], Protein 3a also displayed ion channel function (Fig. 4b). Thus, SARS-CoV-2 contains not only two, but four viroporins as putative drug targets for future therapeutics. Inspired by this, we used the *X. laevis* oocyte expression platform to test a selected set of compounds previously established for their action as ion channel inhibitors.

**All SARS-CoV-2 viroporins can be inhibited by at least one ion channel blocker.** The series of drugs tested on Protein E from SARS-CoV-2 were tested also on the viroporins Protein 3a, ORF7b, and ORF10 by monitoring the ion channel activity in the presence of 10 μM of each drug (Fig. 5). Four of these, rimantadine, adamantane, pyronin B, and -Y had no effect on any of the three viroporins, while xanthene, emodin, and amantadine attenuated at least two out of three ion channels (Fig. 5 and Supplementary Table 1). Current activity facilitated by Protein 3a

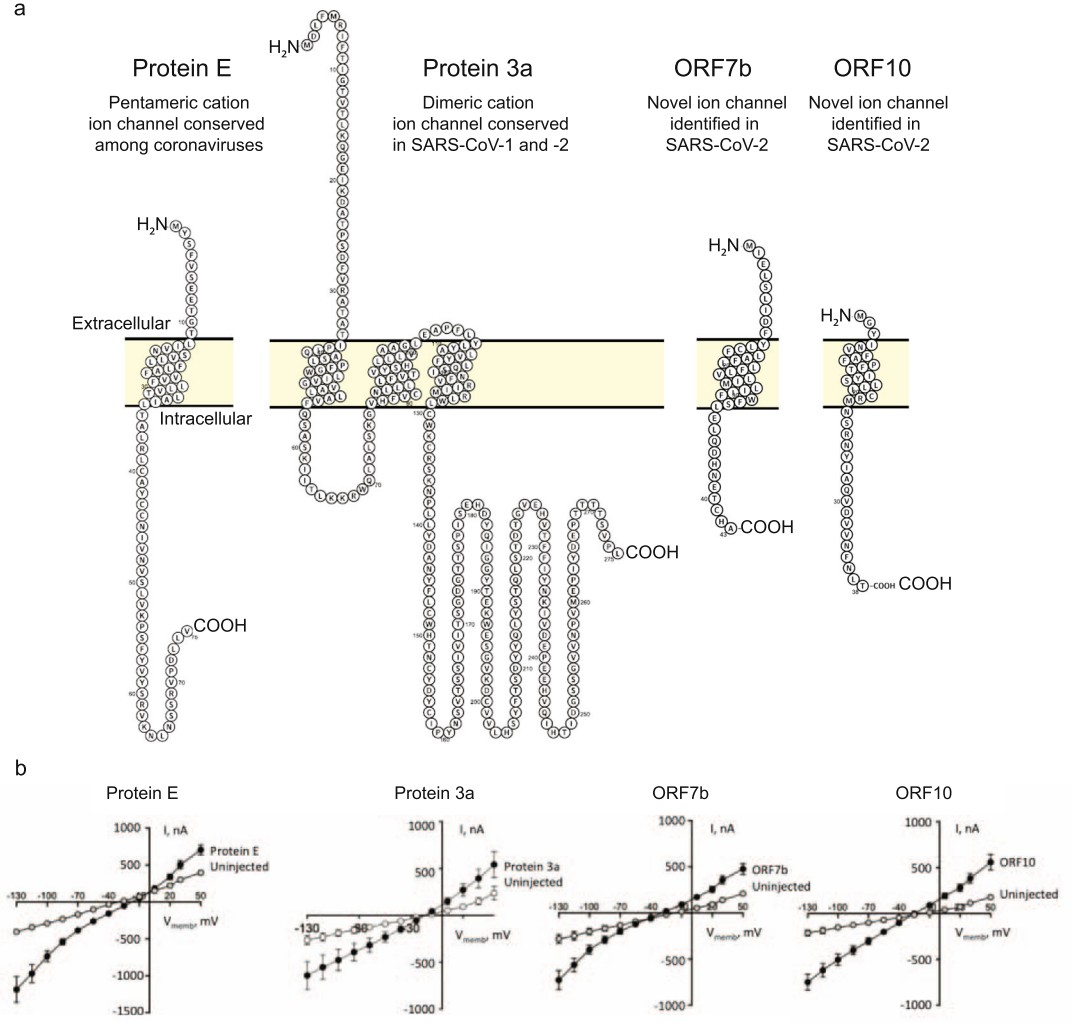

**Fig. 4 Overview of four viroporins encoded by SARS-CoV-2: Protein E, Protein 3a, ORF7b, and ORF10. a** Overall structure and ion channel function of known (Proteins E and 3a) and novel viroporins identified in the current study (ORF7b and ORF10). The membrane topology was predicted by TMHMM2[67] and displayed using Protter[68]. **b** Ion channel activity mediated by Protein E, Protein 3a, ORF7b, and ORF10 electrophysiologically monitored in *X. laevis* oocytes, here shown as summarized and averaged I/V relations in oocytes expressing the viroporins compared to control (uninjected) oocytes of $n = 3$ biologically independent experiments.

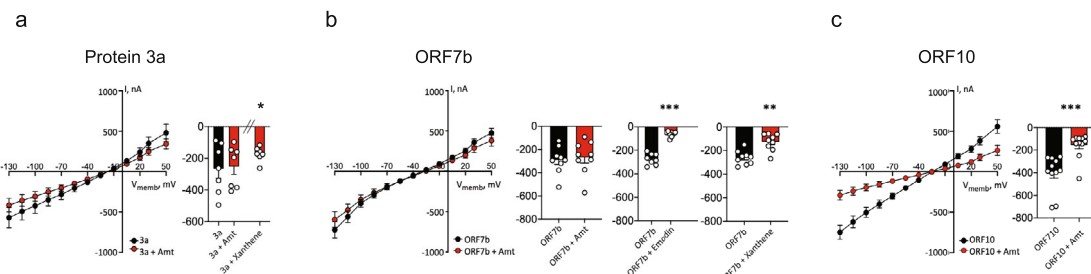

**Fig. 5 All four viroporins of SARS-CoV-2 can be blocked by drugs. a** Protein 3a-, **b** ORF7b-, and **c** ORF10-mediated current activity was monitored in the absence and presence of selected drugs (10 μM). Amantadine was ineffective on Protein 3a (**a**) and ORF7b (**b**), but blocked the activity of ORF10 (**c**). Xanthene blocked the activity of both Protein 3a (**a**) and ORF7b (**b**), while emodin blocked ORF7b (**c**). **a–c** Current activity at −85 mV with the treatment of the specific drugs normalized to the current activity obtained without treatment. Statistical significance was determined with unpaired Student *t* test, **$P < 0.01$; ***$P < 0.001$, of $n = 3$ biologically independent experiments.

was modestly blocked by xanthene (20%, $p = 0.011$, Fig. 5a), while xanthene and emodin reduced the ORF7b-mediated current activity by 47% ($p = 0.001$) and 79% ($p = 0.0005$), respectively (Fig. 5b). The lack of activity of emodin on Protein 3a was a surprise, given the previous inhibition of the homologous Protein 3a from SARS-CoV-1[34], which we confirmed in our experimental setup (Supplementary Fig. 5a). Likewise, emodin revealed no inhibition of ORF10 (Supplementary Fig. 5c). Amantadine was also tested on Protein 3a, ORF7b, and ORF10, of which only ORF10 expressing oocytes were significantly reduced (61% reduction in current, $p = 0.0002$, Fig. 5a–c, respectively). Taken together, amantadine blocked two of the four ion channels in SARS-CoV-2 (Protein E and ORF10) and the activity of all four ion channels in SARS-CoV-2 was significantly inhibited by at least one of the eight applied drugs.

## Discussion

Here, we show that amantadine and HMA block Protein E-mediated current and we provide a likely sequence-specific interaction profile using solution NMR and MD simulations. Our data suggest that polar elements in these drugs can be hydrogen bonded to N15 with their lipophilic group stabilized inward in the channel pore close to L18. Of note is that both compounds also perturb the C-end of Protein E and this effect warrants future studies in correlation with electrophysiology results. Amantadine and HMA are known ion channel blockers used for other purposes in the clinic. This is highly relevant for the current COVID-19 pandemic, as inhibition of Protein E ion channel activity is likely to inhibit both viral replication and virus-mediated inflammation[18,19]—a central component in COVID-19 pathogenesis and morbidity. Of note, amantadine has already been used in the clinic for >45 years for the treatment of influenza A infections and Parkinson's disease, and has also been employed in pregnant women and children suffering from neuropsychiatric disorders. Amantadine may therefore also be suitable for the treatment of diverse groups of patients in future COVID-19 treatment regimens. Ongoing clinical studies carried out by us and other groups (https://clinicaltrials.gov) will reveal the potential of amantadine for the treatment of COVID-19.

Consistent with our results, SARS-CoV-2 Protein E ion channel activity was recently established in a setup similar to ours where it was found to be located at the endoplasmic reticulum–Golgi intermediate complex, increasing the pH of this organelle and to be permeable to monovalent cations such as $Na^+$, $Cs^+$, and $K^+$[44]. Building on these results, we show here that SARS-CoV-2 contains an additional three genes (Protein 3a, ORF7b, and ORF10), encoding membrane proteins with ion channel activity (viroporins), and that not only Protein E, but also Protein 3a, ORF7b, and ORF10 can be inhibited by known ion channel blockers. These findings not only represent a unique opportunity to further study and delineate the viroporins' contribution to SARS-CoV-2 biology and their potential role in pathogenesis and disease progression, but importantly also create the basis for further anti-SARS-CoV-2 drug development.

A remarkable outcome of our electrophysiology experiments was that rimantadine, in contrast to amantadine and HMA, did not block SARS-CoV-2 Protein E-mediated current. This is surprising, given the inhibition by rimantadine of the homologous Protein E from SARS-CoV-1 shown both here and earlier. Our functional data are however consistent with the small CSPs in the solution NMR spectrum of the SARS-CoV-2 full-length Protein E for rimantadine compared to HMA and amantadine. Similarly, a less stable interaction was observed in the complex of rimantadine with Protein E in MD simulations, compared with complexes with

HMA and amantadine. Thus, a subtle change in adamantane drug structure can result in significantly different functional potency. The sequences of Protein E from SARS-CoV-2 and -1 are identical in the transmembrane regions, and the only differences are found in four residues in the C-termini (Supplementary Fig. 2a and b). Mutational studies of these residues and a correlation to the binding and impact of rimantadine and amantadine warrant further investigation.

Another interesting outcome was the observation that the SARS-CoV-2 Protein 3a was not inhibited by emodin, in contrast to its action on the SARS-CoV-1 homolog[34]. However, in support of our data, a recent cryo-EM structure of SARS-CoV-2 Protein 3a, solved in the presence of 100 μM emodin, did not reveal any bound emodin and no structural changes were observed compared to the apo-protein structure in the presence of emodin[25]. Moreover, consistent with our data, it was reported that emodin could not inhibit the ion conductance of SARS-CoV-2 Protein 3a[34].

Although ORF7b functioned as an ion channel and showed some homology to Protein E, we observed no effect of amantadine on ORF7b. The suggested pentameric model for ORF7b[40] is based on similarity to leucine zipper proteins and not homology to Protein E. Based on our sequence alignment, N15 in Protein E, which we here suggested interacting with the ammonium group of amantadine or the guanidinium of HMA with the drugs in the outward orientation, is equivalent to D8 in ORF7b. This aspartic acid residue can form ionic hydrogen bonding interactions with the ammonium group of amantadine or guanidinium group of HMA. Moreover, the Protein E residues (L18 and L21), suggested to interact with the adamantyl group of amantadine or the azepinyl group of HMA in the outward orientation, are conserved in ORF7b. In the ORF7b model, D8 is positioned in the pentamer helices interface and does not point to the pore of the channel. Nevertheless a slight twist of the helices would bring it into the pore in a position comparable to N15 for SARS-CoV-2 Protein E. L18 and L21 in the ORF7b model are oriented into the pore similarly to these residues of SARS-CoV-2 protein E, thereby it is surprising that amantadine did not block ORF7b. It was however inhibited by emodin and xanthene in our study, two similarly sized but very distinct molecules with respect to hydrophobicity. The data at hand warrants further investigations into the actual structure and function of the ORF7b protein and its potential inhibitors.

Overexpression of ORF10 has been shown in severe cases of COVID-19, whereas in milder cases its expression seems to be minimal[45]. Furthermore, genome analyses of SARS-CoV-2 have revealed a reduced level of mutational frequencies in ORF7b and ORF10, compared with other genes in this virus, suggesting functions directly associated with viral fitness, and thereby rendering these potentially interesting therapeutic targets for controlling the COVID-19 pandemic[46,47]. The amino acids suggested to stabilize amantadine in the outward orientation in Protein E (N15, L18, and L21) could be speculated to correspond to ORF10 residues N5, A8, and F11, respectively. Further electrophysiology studies with the inhibitors identified here using Protein E N15 mutants along with mutations of D8 in ORF7b and N5 in ORF10 would be interesting with respect to further exploration of drug sensitivity. With a potential dual inhibitory effect of amantadine on SARS-CoV-2 through inhibition of both Protein E and of ORF10, amantadine stands out as a strong therapeutic opportunity, which may even be less prone to resistance development due to its dual-targeting mechanism.

Owing to the dual inhibitory effect of amantadine on Protein E and ORF10 from SARS-CoV-2 and the inhibition of the former by additionally HMA, emodin, and xanthene, an

inhibitory mechanism of action beyond blocking the viroporins should not be ruled out. It is noteworthy that the solution NMR spectra of the full-length-E protein displayed perturbations upon addition of either amantadine or HMA for transmembrane residues, but also residues towards the C-end of the E protein. This potential for binding promiscuity is reflected in the broad range and overlap of viroporins inhibited by amantadine. It has previously been shown that amantadine binds and blocks influenza A M2[9,48,49] and inhibits the conductance of SARS-CoV-1 Protein E[14], HCV p7[50], Dengue M[51], and chikungunya virus 6K[52], and that HMA inhibits HCV p7, Dengue M, and HIV-1 vpu. Amantadine has also been shown to interact with the HCV p7 in an NMR structure[53] where both amantadine and rimantadine were suggested to act allosterically, inhibiting channel opening and thereby cation conductance. Despite these structural data, amantadine has also been reported not to inhibit p7 ion channel activity[54]. Similar binding promiscuity appears to exist for emodin and xanthene. Emodin has been observed to block the SARS-CoV-1 spike protein-ACE2 interaction and hence it appears to potentially have multiple routes to reduce COVID-19, if it also targets this interaction in SARS-CoV-2[55]. Similar complex action has been described in influenza A virus, where emodin has been shown to inhibit virus replication and viral pneumonia in mice by alterations of intracellular signaling pathways[56].

The direct need for therapies against SARS-CoV-2 infections is obvious and inspires strategies of repurposing drugs approved for other indications, e.g., remdesivir (originally developed for ebola treatment) and steroids (anti-inflammatory treatment)[4]. We propose to further test amantadine as a novel and effective way to treat COVID-19 through its ability to inhibit known (Protein E) and novel (ORF10) ion channels. In addition, amantadine variants, especially those with hydrophobic adducts, can act as lysosomotropic drugs[57], which accumulate in intracellular vesicles through membrane permeation by the electroneutral form and increase intravesicular pH, causing endosome and/or trans-Golgi network neutralization and inhibition of viral reproduction[58]. In support of an antiviral effect, amantadine was recently shown to inhibit SARS-CoV-2 replication in vitro in Vero 6 cells[6]. Moreover, based on a questionnaire, amantadine appeared to be protective for COVID-19 manifestation among 22 patients suffering from Parkinson's disease, multiple sclerosis, or cognitive impairment[4]. To finally establish a beneficial effect of amantadine on COVID-19 treatment, randomized, placebo-controlled, double-blinded studies are needed. At present, amantadine is included in two such studies worldwide (http://clinicaltrials.gov; NCT04794088 and NCT04854759).

The manufacture of amantadine is uncomplicated and cheap and a distribution system is already in place, making the drug readily available for the global community. Globally, >6000 clinical investigations (www.clinicaltrials.gov) have been initiated, testing a wide variety of approaches to prevent, treat, relieve, and diagnose SARS-CoV-2 infection. Although this is indeed impressive and innovative from a scientific, developmental, and medical point of view, it also illustrates an unprecedented open and collaborative approach by regulatory authorities worldwide—underlining the enormous and urgent medical need. We, therefore, propose that amantadine could be an efficient, cheap, and readily available treatment of COVID-19, possibly in combination with other antivirals and/or anti-inflammatory drugs, warranting testing for this purpose, better today than tomorrow. The use of amantadine rests on several decades of clinical experience, facilitating a potential repurposing for the prevention and treatment of SARS-CoV-2 infection and pathogenesis.

## Methods

**RNA preparation and heterologous expression in *X. laevis* oocytes**. The viroporin gene constructs were cloned into the pXOOM vector[59] between the *Bam*HI and *Not*I restriction sites with a Kozak sequence (5′-ACC<u>ATG</u>-3′, initiator ATG underlined) following the BamHI site. Gene synthesis and cloning was performed at GenScript (USA). The plasmids were transformed into *Escherichia coli* TOP10 cells according to the manual. Cells were plated on LB-agar plates with 50 µg/mL Kanamycin and incubated at 37 °C overnight. A single colony from the plate was used to inoculate 75 mL of LB broth containing 50 µg/mL Kanamycin in a 250 mL glass baffled shake flask in a shaking incubator (250 rpm) overnight at 37 °C. Fifty mL of the overnight culture was Midiprepped (GenElute™ HP Plasmid Midiprep Kit—Sigma) according to the manual. The concentration of the isolated plasmids was determined by absorbance ($A_{260 nm}$). The plasmids were all precipitated with isopropanol according to the instructions in the Midiprep manual and dissolved in a smaller volume and stored at −80 °C. The plasmids were linearized downstream from the poly-A segment with XbaI and purified with the High Pure PCR Product Purification Kit (Roche) and their concentration was determined by absorbance ($A_{260 nm}$). The linearized plasmids were in vitro transcribed using T7 mMessage machine according to the manufacturer's instructions (Ambion, Austin, TX). mRNA was extracted with MEGAclear (Ambion, Austin, TX). The concentration of the purified mRNA was determined by absorbance ($A_{260 nm}$) and microinjected into defollicated *X. laevis* oocytes. Oocytes were either purchased from Ecocyte Bioscience, Germany, or surgically removed (in house) from the *X. laevis* frogs. The follicular membrane was removed by incubation in Kulori medium (90 mM NaCl, 1 mM KCl, 1 mM CaCl₂,1 mM MgCl₂, 5 mM HEPES, pH 7.4, 182 mOsm) containing 10 mg/ml collagenase (type 1; Worthington, NJ, USA) and trypsin inhibitor (1 mg/ml; Sigma, Denmark) for 1 hour. Subsequently, the oocytes were washed five times in Kulori medium containing 0.1% bovine serum albumin (Sigma) and incubated in 100 mM $K_2HPO_4$ with 0.1% BSA for 1 h[60]. Animal handling was performed under a license from the Danish Ministry of Justice and in agreement with the European Community guidelines for the use of experimental animals. Oocytes were kept in Kulori medium for 3–4 days at 19 °C prior to experiments.

**Electrophysiology in viroporin-expressing *X. laevis* oocytes**. Oocyte electrophysiology was performed with TEVC at room temperature using borosilicate glass capillary electrodes with a resistance of 1–3 MΩ when filled with 1 M KCl. The conductance was measured using the pClamp 9.2 or 10.4 software (Axon Instruments, Molecular Devices, San Jose, US) together with the Clampator One amplifier (model CA-1B, Dagan Co., Minneapolis, US) and the A/D converter Digidata 1440 A (Molecular Devices, San Jose, US). The currents were low-pass filtered at 500 Hz and sampled at 1 kHz. All current measurements were derived from a 13-step voltage clamp protocol (200 ms, 15 mV increments from −130 mV to +50 mV) with a holding potential of −20 mV. The control solution contained the following in mM: 100 NaCl, 2 KCl, 1 CaCl₂, 1 MgCl₂, 10 HEPES, adjusted to a pH of 7.4 with 2 M tris-base. In experiments employing inhibitors, the oocytes were locally perfused via the recording chamber prior to and during recordings. All inhibitors were purchased from Sigma and dissolved either in MilliQ (Amantadine, CAS no 665-66-7, SMILES: C1C2CC3CC1CC(C2)(C3)N; Pyronin B, CAS no 2150-48-3, SMILES: CCN(CC)C1=CC2=C(C=C1)C=C3C=CC(=[N+](CC)CC)C=C3O2.[Cl−]; Pyronin Y, CAS no 92-32-0), SMILES: CN(C)C1=CC2=C(C=C1)C=C3C=CC(=[N+](C)C)C=C3O2.[Cl−] or dimethyl sulfoxide (DMSO) (Rimantadine, CAS no 1501-84-4, SMILES: CC(C12CC3CC(C1)CC(C3)C2)N; Adamantane CAS no 281-23-2, SMILES: C1C2CC3CC1CC(C2)C3; Emodin, CAS no 518-82-1, SMILES: CC1=CC2=C(C(=C1)O)C(=O)C3=C(C2=O)C=C(C=C3O)O; Xanthene, CAS no 92-83-1, SMILES: C1C2=CC=CC=C2OC3=CC=CC=C31; HMA, CAS no 1428-95-1, SMILES: C1CCCN(CC1)C2=NC(=C(N=C2Cl)C(=O)N=C(N)N)N) and diluted in control solution to a final test concentration of 10 µM. The individual drugs were diluted in MilliQ or DMSO on the day of the experiments, and the corresponding vehicle controls were applied. When applying inhibitors dissolved in DMSO, the same concentration was added to the control solution). Original stocks were stored at room temperature (kept dark). Employed working solutions were discarded after each experimentation. A minimum of nine oocytes were tested per group.

**Solution NMR spectroscopy**. Protein E with all cysteines in its sequence mutated to alanine was expressed from a modified pET15 (Novagen) fusion protein construct as inclusion bodies in the *E. coli* strain BL21(DE3). To produce perdeuterated ${}^{15}$N, ${}^{13}$C labeled protein, the expression was performed in minimal medium with water replaced by D2O (Eurisotope) and supplemented with ${}^{15}$N NH₄Cl (Sigma-Aldrich) as nitrogen source and deuterated ${}^{13}$C-D-glucose (Sigma-Aldrich) as carbon source. The protein was solubilized from the inclusion bodies with 20 mM Tris, pH 8.0, 300 mM NaCl, 1% Fos-Choline-16 (Anatrace), 10 % (w/v) glycerol, Complete-EDTA™ (Roche), 0.5 mM PMSF (Roth, Germany) and purified by metal affinity chromatography (Ni-NTA Agarose, Macherey Nagel, Germany). The thrombin-cleaved protein was purified by reversed-phase HPLC, lyophilized, and refolded at 200 µM final protein concentration in NMR buffer (50 mM NaH2PO4, pH 6.5, 50 mM NaCl, 40 mM Fos-Choline-16, 7.5% D2O, 0.5 mM Pefabloc (Roth, Germany)). The protein E:drug molar ratio used in the samples was 1:10, i.e. 2 mM, resulting in a stoichiometry of 1:50 with respect to pentamers. The resulting sequence of Protein E, including 3 residues from the

tag:GSHMYSFVSEETGTLIVNSVLLFLAFVVFLLVTLAILTALRLAAYAANIV NVSLVKPSFYVYSRVKNLNSSRVPDLLV. All NMR spectra tracking CSPs from small molecules were performed in a 600 MHz solution NMR spectrometer equipped with a prodigy probe using 3 mm NMR tubes filled to 130 μL. For assignments, a 600 MHz spectrometer equipped with a cryoProbe was used. Rimantadine, amantadine, and HMA were purchased from Sigma-Aldrich and used at a concentration of 2 mM. DMSO was used to solubilize HMA, and CSP was calculated against an apo sample with DMSO (2% in the final sample). The DMSO did not produce any significant changes in the spectrum.

**Biomolecular simulations.** The N- and C-termini of the Protein E (aa 8–65; PDB ID 5X29[16]) were capped by acetyl and methylamino groups, respectively, after applying the protein preparation module of Maestro[61]. The protein–drug complexes from docking calculations (see Supporting Information) were embedded in a pre-equilibrated, hydrated POPC lipid bilayer extending 20 Å beyond the solute in the x-y plane with ions, having an orientation with respect to the membrane plane (x-y plane) suggested by the "Orientations of Proteins in Membranes" server[62,63], and a 20 Å layer of waters in the z axis. The proteins and lipid systems were solvated using the TIP3P water model[64]. Each complex was placed in an orthorhombic periodic box with dimensions ($100 \times 96 \times 107$ Å$^3$) for Protein E (aa 8–65). Na$^+$ and Cl$^-$ were placed in the aqueous phase to neutralize the systems and reach the experimental salt concentration of 0.150 M NaCl. Membrane generation and system solvation were conducted with the "System Builder" utility of Desmond[65]. The 20 Å POPC lipid buffer consists of ca. 278 lipids, 22,500 TIP3P water molecules, 78 Cl$^-$ and 62 Na$^+$ ions and the total number of atoms in each system was ~105,000. The stability of the Protein E-complexes was investigated at 310 K using 100 ns-MD simulations at the NPT ensemble with Desmond software. The OPLS2005[66] force field was used for the protein and lipids and intermolecular interactions and the GAFF[63] force field parameters for the ligands. Ligand electrostatic parameters were calculated with the ANTECHAMBER module of Amber14 (DOI: 10.13140/RG.2.2.17892.37766). A force constant of 2 kcal mol Å$^{-2}$ was applied to the Ca carbons of Protein E (aa 8–65). In total three MD simulation repeats were performed using the same starting structure with each simulation performed with randomized velocities. Details for the MD simulations protocols and analysis can be found in the Supporting Information. All the MD simulations were run on ARIS and CyTERA Supercomputing Systems or workstations using the GPU implementation of the MD simulations codes.

**Statistics and reproducibility.** To evaluate statistically significant differences between mean values of two groups we employed an unpaired Student's t test. P values of $p < 0.05$ were considered statistically significant. All statistical analyses were performed in GraphPad Prism 8.0 (GraphPad Software, Inc.) and indicated in the respective figure legend. All data are given as mean values ± standard error of the mean with at least three independent donor frogs employed for the X. laevis oocytes (with a minimum of nine oocytes in total) experiments.

## Data availability

All data generated or analyzed during this study are included in this published article and its supplementary information files. Sequences of used plasmids are available on request. The source data for the graphs and charts in the main manuscript file is available as Supplementary Data 1 and any remaining information can be obtained from the corresponding author upon reasonable request

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

## Acknowledgements

We thank Julius Maximilian Knerr for excellent technical assistance with the drug handling for the electrophysiological studies, Dr. P. Lagarias for preliminary MD simulations related to this project. The NMR studies were supported by Max Planck Society. We moreover thank Dr. Vytautas Gappsys and professor Bert de Groot from Max Planck Institute for Biophysical Chemistry, Goettingen for fruitful discussions during the experiments. The study was supported by grants from the NovoNordisk Foundation (NF20OC0062899), the European Research Council, ERC, (CoG, 682549) and the Lundbeck Foundation (R268-2017-409) to M.M.R., and an Emmy Noether Grant (DFG Grant (AN1316)) to L.B.A.

## Author contributions

The experiments were carried out as follows: T.L.T.B., A.M., and B.H.B. did the electrophysiological measurements, E.T. and A.K. did the molecular simulations, K.X., K.G., S.B., and L.B.A. did the NMR analyses, M.G.J. did the bioinformatics analysis of the viral genomes. M.M.R., T.N.K., and M.G.J. conceived the idea and provided the funding. M.M.R., T.L.T.B., and M.G.J. wrote the first draft of the manuscript with input from all authors.

## Competing interests

The authors declare that there are no competing interests in this work. All authors contributed to the writing of the manuscript and are accountable for all aspects of the work and all persons designated as authors qualify for the authorship, and all those who qualify for authorship are listed. All authors have read and approved the final version of the manuscript.
