## [Peer Review File · Communications Biology]

Reviewers' comments:

Reviewer #1 (Remarks to the Author):

The manuscript describes a study on identification of new ion channels in SARS COV-2 genome , as well as testing some known ion channel blockers (some of them approved drugs) as inhibitors of those channels as potential COVID-19 therapies.

The paper results seems interesting - it is written pretty clearly. However the results are somewhat contradicting to existing studies- so it would be good to elaborate it further.

The main story is that approved flu drug amantadine is inhibiting Protein E for both Sars Cov and Sars cov2 (which is well established in multiple publications), but also targets Orf10 the protein of previously unclear function, which authors propose to be ion channel. Although this sounds very interesting - there are papers discussing that orf10 in sars cov2 may be non coding both in vitro and in vivo so it would be good to discuss and perhaps provide more evidence/information on this

Regarding the other compounds - emodine according to literature is blocking protein 3a (at least in sars cov) , however the authors don't see this in their experiments for SARS-COV2 - this would be good to address. Perhaps show the difference between SARS-COV and SARS COV2.

In the literature there is some experimental data on binding sites for amantadine in protein E - it would be nice to model/ validate(perhaps with some mutation studies) binding site information for other active compounds for the future drug discovery studies, not just for Protein E but for other channels too.

Finally it would be nice to put all the proposed sequence alignments between the channels as well as molecular information (smiles) of the compounds in the supplement.

Reviewer #2 (Remarks to the Author):

The manuscript titled "Amantadin has potential for the treatment of COVID-19 because it targets known and novel ion channels encoded by SARS-CoV-2" is an attempt to repurpose the anti-influenza drug amantadine and its derivatives for the treatment of SARS-CoV-2. Amantadine is known to block the M2 ion channel of Influenza Virus, although the usage of amantadine for treatment of Influenza has been discontinued in various places due to development of resistance in strains. The authors have shown that Amantadine, Emodine and Xanthene show significant blockage of ion -channels formed by SARS-CoV-2 proteins E, Orf7b and Orf10 in *Xenopus leavis* oocytes using standard electrophysiology methods. As ion-channel proteins are crucial for the assembly and pathophysiology of SARS-CoV-2, as well as for other enveloped viruses, it is a possibility that amantadine may be repurposed for SARS-CoV-2 treatment.

The main deficiency of the work is lack of any virus infectivity data. Although the ion channel blockage by amantadine and other derivatives seems promising, it is necessary that the effect of these molecules on SARS-CoV-2 replication in cell culture is tested at various concentrations, to substantiate the claims made in the manuscript.

Also, since the experimental structure of SARS-CoV-2 E protein is currently available (and cited in the manuscript), it will be helpful to understand the molecular mechanism for amantadine binding to the channel by structural bioinformatics based analysis. The complete lack of effect of close derivatives like rimantadine and adamantane on ion channel activity of E is surprising, and a molecular level explanation is necessary in this context.

The effect of amantadine on ion channel proteins of other enveloped viruses like HCV and Chikungunya (Griffin et al., FEBS Lett, 2003; Dey et al., PLoS Negl Trop Dis, 2019) and the consequent effect on viral replication (Dey et al., PLoS Negl Trop Dis, 2019) is established in literature, and should be cited in the manuscript.

The manuscript may be considered for publication upon resolution of the major issues as mentioned.

Dear Reviewers

We hereby resubmit the manuscript “Amantadine has potential for the treatment of COVID-19 because it targets known and novel ion channels encoded by SARS-CoV-2”, with the internal track number COMMSBIO-20-3500-T, to Communications Biology.

The idea of using amantadine as treatment for COVID-19, as concluded in the first submission, have received much supporting evidence by data published after our first submission. It has thus been shown that:

- 1) Amantadine inhibits SARS-CoV-2-replication in vitro (Fink et al., 2021)
- 2) Moreover, data from a retrospective clinical cohort study in Mexico (with ~160.000 persons), revealed apparent protection of amantadine in COVID-19 patients (Mancilla-Galindo et al., 2021)
- 3) In addition, clinical data from a small scale treatment of 15 COVID-19 patients with amantadine showed promising effects (Aranda-Abreu et al., 2020)
- 4) Finally, data based on a questionnaire revealed a protective role of amantadine, used in Parkinson’s disease, multiple sclerosis or cognitive impairment, for COVID-19 manifestation (Rejdak & Grieb, 2020)

Thus, from being a hypothesis, driven by our molecular uncovering of amantadine as inhibitor of one known (Protein E) and one novel viroporin (ORF10) from SARS CoV-2, we now have strong clinical evidence that this is in fact a potential treatment of COVID-19 patients.

Below you will find a point-to-point reply to your comments and requests. Our replies are written in blue and in italic letters. As an initial remark, we would like to thank you for your valuable input and suggestions for changes and addition experiments. We believe that the resubmitted manuscript is improved by the additional experiments. To perform some of the requested experiments, we have teamed up with researchers behind the article in Nat. Struct. Biol. describing the structure of the transmembrane part of Protein E (Mandala et al., 2020) and performed molecular dynamics simulations as well as solution NMR in micelles for structural investigations of full-length protein E SARS-CoV2 in complex with amantadine and rimantadine as well as in complex with hexamethylene-amiloride (HMA).

A brief list of extra experiments (described more comprehensively below in the point-to-point response) includes:

1. Additional electrophysiological testing of ion channel blockers on Protein E and Protein 3a from SARS CoV-1.
2. Inclusion of additional ion channel blockers in the electrophysiological testing on Protein E from SARS CoV-1 and -2.
3. Solution NMR in micelles studies of samples of the full-length protein E SARS-CoV2 with ligands to describe binding modes of amantadine, rimantadine and HMA on Protein E from SARS-CoV-2.
4. Molecular dynamics simulations to suggest the binding profile of amantadine, rimantadine and HMA in complex with Protein E from SARS-CoV-2.

Point-to-point response:

Reviewer #1 (Remarks to the Author):

The manuscript describes a study on identification of new ion channels in SARS COV-2 genome , as well as testing some known ion channel blockers (some of them approved drugs) as inhibitors of those channels as potential COVID-19 therapies.

Issue 1) The paper results seems interesting - it is written pretty clearly. However the results are somewhat contradicting to existing studies- so it would be good to elaborate it further.

Reply: We agree with you in this aspect, and have performed additional experiments to clarify this difference (explained in details below).

Issue 2) The main story is that approved flu drug amantadine is inhibiting Protein E for both Sars Cov and Sars cov2 (which is well established in multiple publications), but also targets Orf10 the protein of previously unclear function, which authors propose to be ion channel. Although this sounds very interesting - there are papers discussing that orf10 in sars cov2 may be non coding both in vitro and in vivo so it would be good to discuss and perhaps provide more evidence/information on this

Reply: we have expanded our Introduction and Discussion regarding the knowledge of ORF10.

Issue 3) Regarding the other compounds - emodin according to literature is blocking protein 3a (at least in sars cov-1), however the authors don't see this in their experiments for SARS-COV2 - this would be good to address. Perhaps show the difference between SARS-COV and SARS COV2.

Reply: In a recent paper, submitted to bioRxiv paper entitled "Cryo-EM structure of the SARS-CoV-2 3a ion channel in lipid nanodiscs" by Kern et al (<https://doi.org/10.1101/2020.06.17.156554>), it was attempted to solve the structure of ion channel protein 3a in the presence of 100 μ M emodin. The authors found no density for emodin and there were no structural changes between protein 3a and the one with emodin. Moreover, in the electrophysiology (EP) experiments there was no blocking in 3a-mediated cation current with emodin. Thus, this paper confirms our findings, which we discuss in the revised manuscript. Moreover, we performed extra experiments and confirmed that emodin was able to block the current of Protein 3a from SARS-CoV-1 in our system, as a positive control. These data are not shown in the manuscript, but inserted below. If you find that these data should be included in the manuscript, we will do so promptly.

Data not shown in the manuscript. Control data for the impact of emodin in Protein 3a from SARS-CoV-1.

Impact of emodin on the ion channel function of Protein 3a from SARS-CoV-1 | A, The current activity at -85mV in SARS-CoV-1 Protein 3a-expressing oocytes normalized to that of uninjected oocytes (red) B, The current activity at -85mV with emodin treatment normalized to the current activity obtained without treatment. C, Summarized and averaged I/V relations in SARS-CoV-1 3a-expressing oocytes in the absence (black) and presence (red) of emodin. N=16. Statistical significance was determined with unpaired Student *t* test, * P < 0.05; **P < 0.01; ***P < 0.001.

Issue 4) In the literature there is some experimental data on binding sites for amantadine in protein E - it would be nice to model/ validate (perhaps with some mutation studies) binding site information for other active compounds for the future drug discovery studies, not just for Protein E, but for other channels too.

Reply: In order to study the effect of amantadine, rimantadine and HMA against E Protein, we have applied (a) solution NMR spectroscopy of the full-length Protein E (1-75) in micelles and (b) MD simulations using the experimental structure with PDB ID 5X29 of E (8-65) resolved by solution NMR in micelles (Surya et al., 2018). In this previously published experimental structure, a binding pose of amantadine was included. Our NMR results suggested that amantadine binds to the N-terminal of the Protein likely close to N15 and the MD simulations suggest that amantadine can bind in this region. For the other channels, there is no experimental structure in complex with drug to pursue these studies.

Issue 5) Finally it would be nice to put all the proposed sequence alignments between the channels as well as molecular information (smiles) of the compounds in the supplement.

Reply: We thank you for this suggestion, and have now included this in the supplemental material.

Reviewer #2 (Remarks to the Author):

The manuscript titled “Amantadin has potential for the treatment of COVID-19 because it targets known and novel ion channels encoded by SARS-CoV-2” is an attempt to repurpose the anti-influenza drug amantadine and its derivatives for the treatment of SARS-CoV-2. Amantadine is known to block the M2 ion channel of Influenza Virus, although the usage of amantadine for treatment of Influenza has been discontinued in various places due to development of resistance in strains. The authors have shown that Amantadine, Emodine and Xanthene show significant blockage of ion -channels formed by SARS-CoV-2 proteins E, Orf7b and Orf10 in *Xenopus leavis* oocytes using standard electrophysiology methods. As

ion-channel proteins are crucial for the assembly and pathophysiology of SARS-CoV-2, as well as for other enveloped viruses, it is a possibility that amantadine may be repurposed for SARS-CoV-2 treatment.

Issue 1) The main deficiency of the work is lack of any virus infectivity data. Although the ion channel blockage by amantadine and other derivatives seems promising, it is necessary that the effect of these molecules on SARS-CoV-2 replication in cell culture is tested at various concentrations, to substantiate the claims made in the manuscript.

Reply: After we submitted our manuscript, another group have conducted in vitro testing assessing antiviral activity of a series of compounds and shown that amantadine in a dose-dependent manner inhibits the replication of SARS-CoV-2. This reference have now been included in the manuscript (Fink et al., 2021) and their data discussed appropriately.

Issue 2) Also, since the experimental structure of SARS-CoV-2 E protein is currently available (and cited in the manuscript), it will be helpful to understand the molecular mechanism for amantadine binding to the channel by structural bioinformatics based analysis. The complete lack of effect of close derivatives like rimantadine and adamantane on ion channel activity of E is surprising, and a molecular level explanation is necessary in this context.

Reply: In the revised manuscript, we investigate the effect of amantadine and rimantadine against E Protein using a combination of solution NMR spectroscopy of the full-length Protein E (1-75) in micelles and MD simulations using the experimental structure PDB ID 5X29 of E (8-65) from solution NMR in micelles (Surya et al., 2018). From our NMR results, a weak binding of rimantadine to E Protein was observed, and our MD simulations also suggested this weak binding.

Issue 3) The effect of amantadine on ion channel proteins of other enveloped viruses like HCV and Chikungunya (Griffin et al., FEBS Lett, 2003; Dey et al., PLoS Negl Trop Dis, 2019) and the consequent effect on viral replication (Dey et al., PLoS Negl Trop Dis, 2019) is established in literature, and should be cited in the manuscript.

Reply: We have cited these articles in the resubmitted version of our manuscript.

Issue 4) The manuscript may be considered for publication upon resolution of the major issues as mentioned.

Reply: We believe that the revised manuscript is significantly improved in its new form, changed according to the suggested corrections and additional experiments.

In conclusion, we have conducted multiple extra experiments and rewritten the text accordingly resulting in an improved manuscript, which we hope that you will find acceptable for publication in Communications Biology. The manuscript has – in its new form – and with the advancements on the *in vitro* virology and clinical site by other researchers – a broad appeal with implications for millions of people worldwide.

Yours sincerely
Mette Rosenkilde

- Aranda-Abreu, G. E., Aranda-Martinez, J. D., Araujo, R., Hernandez-Aguilar, M. E., Herrera-Covarrubias, D., & Rojas-Duran, F. (2020). Observational study of people infected with SARS-Cov-2, treated with amantadine. *Pharmacol Rep*, 72(6), 1538-1541.
<https://doi.org/10.1007/s43440-020-00168-1>
- Fink, K., Nitsche, A., Neumann, M., Grossegasse, M., Eisele, K. H., & Danysz, W. (2021). Amantadine Inhibits SARS-CoV-2 In Vitro. *Viruses*, 13(4).
<https://doi.org/10.3390/v13040539>
- Mancilla-Galindo, J., Garcia-Mendez, J. O., Marquez-Sanchez, J., Reyes-Casarrubias, R. E., Aguirre-Aguilar, E., Rocha-Gonzalez, H. I., & Kammar-Garcia, A. (2021). All-cause mortality among patients treated with repurposed antivirals and antibiotics for COVID-19 in Mexico City: A real-world observational study. *EXCLI J*, 20, 199-222.
<https://doi.org/10.17179/excli2021-3413>
- Mandala, V. S., McKay, M. J., Shcherbakov, A. A., Dregni, A. J., Kolocouris, A., & Hong, M. (2020). Structure and drug binding of the SARS-CoV-2 envelope protein transmembrane domain in lipid bilayers. *Nat Struct Mol Biol*, 27(12), 1202-1208.
<https://doi.org/10.1038/s41594-020-00536-8>
- Rejdak, K., & Grieb, P. (2020). Adamantanes might be protective from COVID-19 in patients with neurological diseases: multiple sclerosis, parkinsonism and cognitive impairment. *Mult. Scler. Relat Disord*, 42, 102163. [https://doi.org/S2211-0348\(20\)30239-X](https://doi.org/S2211-0348(20)30239-X) [pii];10.1016/j.msard.2020.102163 [doi] (Not in File)
- Surya, W., Li, Y., & Torres, J. (2018). Structural model of the SARS coronavirus E channel in LMPG micelles. *Biochim. Biophys. Acta Biomembr*, 1860(6), 1309-1317.
[https://doi.org/S0005-2736\(18\)30058-0](https://doi.org/S0005-2736(18)30058-0) [pii];10.1016/j.bbamem.2018.02.017 [doi] (Not in File)

REVIEWERS' COMMENTS:

Reviewer #1 (Remarks to the Author):

In the revision the authors addressed my previous comments with new data. I would also suggest to the authors to include SARS-COV2 data for emodine in the supplement as control. Also it would be interesting to run Alphafold (and include the models if they are confident) for new suspected Ion channel in addition to alignments to see if it further supports the model(once model is generated one can run structure similarity like fatcat to see. Otherwise I think the paper can be published

Reviewer #2 (Remarks to the Author):

The resubmitted manuscript has addressed all previous concerns and critiques adequately. A major weakness of the previous manuscript was the lack of any mechanistic explanation of the variation between the binding of amantadine and amantadine-like drugs to E protein. In the revised manuscript, the authors have conducted solution NMR and MD simulation studies to provide mechanistic details of drug binding, which correlates with the electrophysiology studies. Further, the manuscript has been extended to include other ion -channel proteins from SARS-CoV-2 as well. The authors have also cited two recent studies which have reported the effect of amantadine on virus replication in cell culture and in human subjects. The manuscript in its present form is suitable for publication and will make a great addition to the literature on the effect of ion channel blockers in SARS-CoV-2 and related viruses.

Dear Reviewers

We hereby re-submit the manuscript after acceptance “Amantadine has potential for the treatment of COVID-19 because it targets known and novel ion channels encoded by SARS-CoV-2”, with the internal track number COMMSBIO-20-3500-T, to Communications Biology.

Since you had some remarks to the resubmitted version from medio July, we have compiled some answers to those remarks below:

Reviewer #1 (Remarks to the Author):

In the revision the authors addressed my previous comments with new data. I would also suggest to the authors to include SARS-COV2 data for emodine in the supplement as control. Also it would be interesting to run Alphafold (and include the models if they are confident) for new suspected Ion channel in addition to alignments to see if it further supports the model(once model is generated one can run structure similarity like fatcat to see. Otherwise, I think the paper can be published

Our Answer: we thank you for your evaluation. We have now included a novel supplemental figure showing the data requested above (Supplementary Fig. 5). While we appreciate the improved opportunities for accurate prediction of our novel and functional ion channels by employing e.g. Alphafold, we believe this is beyond the scope of this paper. We have chosen to first of all focus on the physiology of Protein E and the ability to employ existing drugs to inhibit the activity of Protein E, and we have provided functional data (including inhibition) on the all the novel ion channels. We have thoroughly re-worked the paper to communicate a coherent and well balanced scientific story, where the functional, structural and model sections each have significant space, details and depth, but importantly also mutually support each other strengthening both the data package as well as the communication of the key messages

Reviewer #2 (Remarks to the Author):

The resubmitted manuscript has addressed all previous concerns and critiques adequately. A major weakness of the previous manuscript was the lack of any mechanistic explanation of the variation between the binding of amantadine and amantadine-like drugs to E protein. In the revised manuscript, the authors have conducted solution NMR and MD simulation studies to provide mechanistic details of drug binding, which correlates with the electrophysiology studies. Further, the manuscript has been extended to include other ion -channel proteins from SARS-CoV-2 as well. The authors have also cited two recent studies which have reported the effect of amantadine on virus replication in cell culture and in human subjects. The manuscript in its present form is suitable for publication and will make a great addition to the literature on the effect of ion channel blockers in SARS-CoV-2 and related viruses.

Our Answer: We thank you for this evaluation.